# SAFD: A STYLE-AGNOSTIC FRAMEWORK FOR DETECTING LLM-GENERATED TEXT

## ABSTRACT

With the rising prominence and fluency of large language models (LLMs), developing technologies to identify LLM-generated text has become increasingly critical. However, existing technologies depend on static linguistic features, which can be evaded as advanced models increasingly mimic a wide range of writing styles. This study reveals two crucial vulnerabilities in existing detection systems:(1) State-of-the-art detectors suffer a substantial accuracy decline, reaching up to 16.45% when exposed to style-based adversarial attacks generated by LLMs. (2) While general-purpose LLMs exhibit remarkable zero-shot capabilities, their performance in detecting adversarially manipulated text is lower than specialized detectors fine-tuned for robustness. To address these vulnerabilities, we propose a novel style-agnostic detection framework named SAFD that enhances detection accuracy and robustness by prioritizing content-driven features over stylistic attributes. Our approach integrates a style-invariant training paradigm to disentangle content semantics from stylistic variations. We leverage adversarially enriched datasets constructed using LLMs fine-tuned for diverse style-based attacks. Furthermore, we utilize advanced representation learning techniques to extract content-centric features, emphasizing semantic coherence, logical consistency, and factual alignment. Experimental results across multiple datasets and detection models validate the effectiveness of our framework, showing improvements in detection accuracy and robustness against diverse adversarial manipulations. The dataset and code are in the link [1].

## 1 INTRODUCTION

Large language models (LLMs) have recently demonstrated remarkable capabilities in generating textual data Zhao et al. (2023); Minaee et al. (2024), and have garnered widespread attention. However, malicious actors abuse LLMs to generate various-style text, creating misleading public-opinion content such as fake news or academic papers with false data to distort the truth Wu et al. (2023). Considering this, it is an extremely pressing matter to develop detection technologies that remain unaffected by text styles and can precisely identify the text's origin.

Many state-of-the-art detection methods heavily rely on stylistic attributes Wu et al. (2023); Wang et al. (2023a); Liu et al. (2023), i.e., lexical choice, sentence structure, and syntactic patterns. Although effective under normal conditions, these features are highly susceptible to adversarial style-based attacks, where generated text mimics human writing styles. This reliance undermines the robustness of these detectors, as adversaries can easily manipulate style while preserving semantic coherence. These limitations highlight the pressing need for detection frameworks that are less reliant on stylistic attributes, incorporate content-focused strategies, and are rigorously evaluated against a comprehensive set of adversarial attacks.

---

[1] https://anonymous.4open.science/status/A-Style-Agnostic-Framework-for-Detecting-LLM-Generated-Text-90B7

In this work, we aim to develop a technique independent of stylistic attributes and unaffected by text styles, enabling precise identification of the text's origin. Specifically, we identify two critical vulnerabilities: (1) *state-of-the-art detectors for generated text are significantly compromised by LLM-induced stylistic variations*, and (2) *general-purpose LLMs exhibit markedly inferior performance in detecting adversarially manipulated text*. Motivated by two critical vulnerabilities, we present a novel style-agnostic detection framework named `SAFD`. We incorporate a style-invariant training paradigm that involves disentangling content features from stylistic attributes during training using adversarial learning techniques. The model is explicitly trained to disregard stylistic variations by introducing adversarially perturbed examples that mimic diverse human writing styles. Then, we generate a comprehensive adversarial dataset leveraging LLMs fine-tuned for style attacks. This dataset includes diverse stylistic manipulations across genres and domains, exposing the model to various adversarial scenarios. Experimental results across various datasets and models demonstrate that our method achieves substantial improvements in detection accuracy. Our contributions are as follows:

- We observe that current detectors for LLM-generated text exhibit considerable limitations when encountering texts with styles purposefully altered by other LLMs. To our knowledge, *this work is the first to systematically investigate the impact of these LLM-driven stylistic alterations on the performance of text detection systems*.

- We introduce a novel training framework that enhances the resilience of text generation detectors by learning style-invariant features. To our knowledge, *this is the first approach to forgo stylistic artifacts in favor of content-driven analysis*, ensuring its broad applicability.

- We construct an adversarially enriched dataset by leveraging LLMs fine-tuned for style-based attacks. *Extensive experiments on advanced and commercial LLMs (LLAMA-13B, GPT-4, Qwen-2, etc.) show that* `SAFD` *outperforms state-of-the-art detection methods by up to* 6.11%.

## 2 RELATED WORK

In this section, we discuss two critical dimensions of LLMs-generated text analysis, i.e., representative detection approaches and adversarial attacks.

**Universal Detection Frameworks for LLM-Generated Content**. Numerous methodologies leverage the sophisticated mechanisms of LLMs, encompassing intermediate layer outputs as well as weight parameters, to differentiate between texts authored by humans and those generated by LLMs Taguchi et al. (2024); Wang et al. (2023b); Bao et al. (2023); Su et al. (2023); Bakhtin et al. (2019). These methodologies introduce subtle text modifications to monitor changes in log probabilities, where a notable decrease typically indicates LLM-generated text, while an increase or minor fluctuations suggest human authorship. To detect text generated by smaller models, statistical analyses are employed to examine features,i.e., word choice, sentence structure, and stylistic elements Taguchi et al. (2024); Gao et al. (2018); Wang et al. (2023b); Shi et al. (2024). These features are compared against known human and LLMs-generated patterns to identify discrepancies indicative of non-human authorship. Despite their innovations, these methods largely depend on stylistic cues, which are easily manipulated by adversaries using style-based transformations Waghela et al. (2024); Fu et al. (2024); Wang et al. (2024a). This reliance exposes a critical vulnerability, as demonstrated by accuracy degradation under adversarial conditions.

**Adversarial Attacks in Text Generation**. Adversarial attacks have now emerged as a formidable threat to the detection of generated text Alzantot et al. (2018); Zhang et al. (2024). The existing adversarial sample attack techniques are characterized by their high level of stealth and aggressiveness, enabling them to effectively evade the current detection methods Huq et al. (2020); Kadhim et al. (2025). Many approaches subtly perturb parts of the text or embed feature-specific text snippets Li et al. (2018); Wang et al. (2019); He et al. (2021); Boreshban et al. (2023), causing NLP models to produce targeted incorrect outputs while

preserving semantic and syntactic integrity. Although adversarial examples in other domains, i.e., image recognition Zhang et al. (2021); Wei et al. (2018); Cui et al. (2024); Navaneet et al. (2024) and fake news detection Zhu et al. (2024); DSouza & French (2024); Wu et al. (2024a), have been extensively studied, similar efforts in the context of LLMs-generated text are limited. Existing detection frameworks are not specifically designed to address these sophisticated style transformations, leading to accuracy degradation in adversarial scenarios.

## 3 PROBLEM DEFINITION

Let $\mathcal{D} = \{(x_i, y_i)\}_{i=1}^N$ represent a dataset where $x_i$ is the input text sample, and $y_i \in \{0, 1\}$ is the corresponding label, with $y_i = 1$ indicating LLMs-generated text and $y_i = 0$ for human-written text. The goal is to train a detector $f_\theta(x) : \mathcal{X} \to \{0, 1\}$ parameterized by $\theta$, which can accurately classify $x_i$ while maintaining robustness against adversarial manipulations.

An adversarial example $x_i'$ is defined as a perturbed version of $x_i$ generated by an adversary $\mathcal{A}$, and $\phi$ represents adversarial parameters.:

$$x_i' = \mathcal{A}(x_i; \phi) \tag{1}$$

The $\mathcal{X}'$ denotes the set of adversarially perturbed samples. . The detector $f_\theta(x)$ should satisfy the following robustness criterion:

$$f_\theta(x_i) = f_\theta(x_i'), \quad \forall x_i \in \mathcal{D}, \ x_i' \in \mathcal{X}', \tag{2}$$

## 4 MOTIVATION

In this section, we employ LLMs to generate texts with diverse stylistic variations. We conduct a preliminary analysis to assess the performance of state-of-the-art detections in identifying this generated content.

### 4.1 DIVERSE STYLISTIC VARIATIONS

The advanced capabilities of LLMs enable users to transform text styles through tailored prompts, challenging the robustness of detection systems against such stylistic alterations. In this study, we investigate a direct style-based attack by employing distinctive writing styles characteristic of texts such as Andersen's fairy tales and the scientific prose found in prestigious academic journals like Nature and Science as prompts. These writing styles are marked by distinctive narrative elements and tonal qualities, making them viable options for adversarial manipulations. For instance, a modern narrative might be rewritten with whimsical language, moral undertones, and vivid imagery characteristic of fairy tales. Our general prompt format for these transformations is structured as follows:

> **Rewrite the following text using the style of [publisher/book]:** [input text]

For narrative texts, we employ the writing style of Andersen's fairy tales to transform stories generated by LLMs. For political texts, we adopt the style of CNN, while for scientific texts, we utilize the writing style of Nature and Science. We employ these stylistically transformed test samples to systematically evaluate the performance of detection systems when subjected to style-oriented adversarial attacks.

### 4.2 STYLE-RELATED DETECTOR VULNERABILITY

As shown in Table 1, the original accuracy (O) reflects the detection methods' performance on unaltered texts, while the adversarial accuracy (A) represents their robustness when faced with style-based adversarial attacks, with a drop in accuracy indicated by (↓). DetectGPT Mitchell et al. (2023) demonstrates the highest original and adversarial accuracies across all datasets, particularly excelling in Story (O: 78.24%, A: 70.13%) and PolitiFact (O: 75.91%, A: 71.96%). However, its performance drops more significantly in the Science dataset (O: 69.72%, A: 60.43%), highlighting the challenge of adversarial at-

| Method | Story | | PolitiFact | | Science | |
|---|---|---|---|---|---|---|
| | **O** | **A** (↓) | **O** | **A** (↓) | **O** | **A** (↓) |
| DetectGPT | 78.24 | 70.13 | 75.91 | 71.96 | 69.72 | 60.43 |
| GLTR | 63.24 | 54.88 | 59.24 | 52.32 | 60.66 | 53.73 |
| LLMDet | 72.35 | 60.69 | 70.08 | 62.57 | 60.52 | 57.63 |
| LLaMA-7B | 69.34 | 61.45 | 73.45 | 60.59 | 68.59 | 52.14 |
| Neo-2.7B | 62.25 | 50.82 | 65.39 | 52.75 | 60.97 | 48.33 |

Figure 1: The performance comparison of different methods across Story, PolitiFact, and Scientists datasets. **O** represents the original accuracy, and **A** (↓) represents the adversarial accuracy.

tacks in more structured or technical domains. GLTR Gehrmann et al. (2019) and LLMDet Wu et al. (2023) exhibit weaker performance, with GLTR demonstrating notable vulnerability under adversarial conditions. For instance, GLTR's accuracy drops on the PolitiFact dataset (O: 59.24%, A: 52.32%). LLMDet performs moderately better, particularly on the PolitiFact dataset (O: 70.08%, A: 62.57%). The LLaMA-7B and Neo-2.7B models exhibit relatively lower original and adversarial accuracies overall, particularly in the Science dataset where Neo-2.7B struggles the most (O: 62.25%, A: 48.33%), underlining the challenges of handling adversarial attacks with smaller or less robust LLMs. Existing detectors and general large language models perform inadequately under adversarial attacks, particularly when handling specialized or technical texts, where their performance degradation is especially pronounced.

*Observation 1 (Style-related vulnerability of LLMs generated text detectors).* State-of-the-art detectors for generated text are found to be impacted by LLM-driven stylistic variations. This impact results in the performance drop, as evidenced by an accuracy decline of up to 16.45% when evaluated on stylistically altered test sets.

*Observation 2 (Limitations of LLMs in text robustness detection).* While LLMs demonstrate remarkable zero-shot capabilities as general-purpose foundational models, their performance in detecting adversarially manipulated text is notably inferior compared to specialized LLMs-generated text detection systems and pre-trained language models fine-tuned for specialized or technical tasks.

## 5 METHODOLOGY

In this section, we propose a method to detect LLM-generated content by transforming human texts into various styles using an LLM for diverse training data. Features combine contextual embeddings and statistical metrics. The training uses three loss functions: style alignment, classification, and pseudo-label supervision. This ensures robustness to stylistic variations and adaptability to different generation patterns.

### 5.1 STYLE-BASED REFRAMING

To simulate the diverse styles that LLM-generated content may exhibit in real-world scenarios, we adopt a data augmentation strategy based on style transformation. Specifically, each human-written text $p_{\text{human}} \in D_{\text{human}}$ is transformed into multiple stylistic variants by leveraging an LLM $M_{\text{LLM}}$. $N_s$ is the number of stylistic transformations, $s_i$ represents a specific style (e.g., "formal," "narrative," or "scientific"), and the equation 3 describes the transformation process.

$$p_{\text{gen},i} = M_{\text{LLM}}(p_{\text{human}}, \text{style} = s_i), i = 1, \ldots, N_s \tag{3}$$

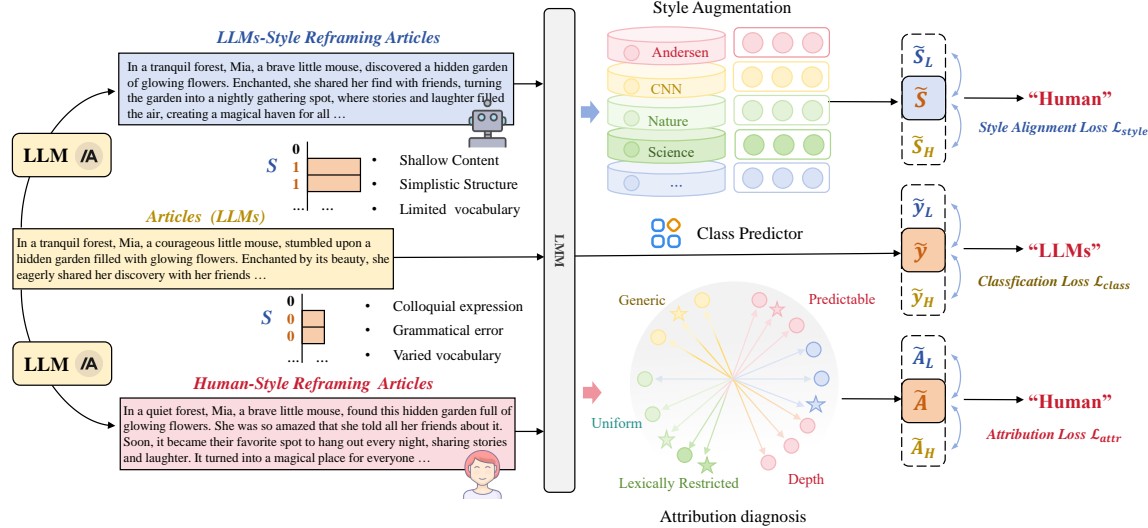

Figure 2: Overview of the SAFD framework. It consists of three main components, e.g., style alignment loss, class detection Loss, veracity attribution loss. SAFD is designed to enhance the robustness of text generation detectors against style-based adversarial attacks by focusing on content-driven features and minimizing reliance on stylistic cues.

The augmented training dataset $D$ is subsequently constructed by integrating stylistic variants derived from human-written texts with equation 4, thereby ensuring a diverse and comprehensive sample set for robust model training.

$$D_{\text{train}} = D_{\text{human}} \cup \{p_{\text{gen},i} \mid i = 1, 2, \ldots, N_s\}. \tag{4}$$

## 5.2 STYLE ALIGNMENT LOSS

To improve robustness to stylistic variations, our style alignment loss enforces consistent predictive distributions across $N_s$ stylistic variants $\{p_{\text{gen},i}\}$. We achieve this by passing the hidden representation $h_{p_{\text{gen},i}}$ of each variant through an MLP classifier $M$ to obtain the output probability distributions:

$$y_i = \text{Softmax}(M(h_{p_{\text{gen},i}})) \tag{5}$$

The style alignment loss $L_{\text{style}}$, is then formulated as the average pairwise Kullback-Leibler (KL) divergence between the predictive distributions of all variant pairs:

$$L_{\text{style}} = \frac{1}{N_s^2} \sum_{i=1}^{N_s} \sum_{j=1}^{N_s} \text{KL}(y_i \parallel y_j) \tag{6}$$

where $\text{KL}(y_i \parallel y_j)$ denotes the KL divergence from $y_j$ to $y_i$ Wu et al. (2024c). This encourages the classifier to produce a consistent output, irrespective of superficial stylistic features.

## 5.3 CLASSIFICATION LOSS

The classification loss ensures that the model correctly predicts the source of the input text. For a given input $p$, with true label $y$ and predicted probabilities $\hat{y}$, the loss is:

$$L_{\text{class}} = -\sum_{k=1}^{K} y_k \log \hat{y}_k \tag{7}$$

where $K = 2$ corresponds to the two classes (human-written and LLMs-generated)

## 5.4 CONTENT-FOCUSED ATTRIBUTION SUPERVISION

The veracity attributions are generated through a process that involves querying a LLM to identify distinctive features and patterns characteristic of text produced by LLMs. We employ the following inquiry method:

> **Input Texts**: [LLMs-generated texts]
> **Question**: Which of the following problems does this text have? Single language style, Too structured logical structure, Lack of background knowledge and personal experience, Repetitive or patterned expression, Data biases, and errors. If multiple options are applicable, provide a comma-separated list ordered from most to least relevant. Answer "No" if none of the options apply.

For a given veracity attributions space $C = \{c_1, c_2, \ldots, c_m\}$, the generated-text $p_{\text{gen},i}$ attribution $A_i$ satisfy:

$$A_i = \begin{cases} 1, & \text{if } p_{\text{gen},i} \text{ satisfies } c_k, \\ 0, & \text{otherwise.} \end{cases} \tag{8}$$

we define a veracity attribution loss based on the binary cross-entropy between the predicted attribution vector $\hat{\mathbf{A}}_i$ and the ground-truth vector $\mathbf{A}_i$. The attribution loss is averaged over $N_a$ samples.

$$L_{\text{attr}} = \frac{1}{N_a} \sum_{i=1}^{N_a} \text{BCE}(A_i, \hat{A}_i) \tag{9}$$

## 5.5 FINAL OBJECTIVE FUNCTION

The overall training objective combines style alignment loss, classification loss, and veracity attribution supervision loss:

$$L = L_{\text{style}} + L_{\text{class}} + L_{\text{attr}} \tag{10}$$

## 6 EXPERIMENTS

In this section, we describe the experimental setup, including dataset details and implementation specifics of SAFD. Eventually, we will assess the performance of our approach, including accuracy and F1 score, etc., on multiple datasets and compare these metrics against state-of-the-art algorithms.

## 6.1 EXPERIMENT SETTING

**Datasets.** Our experiment employs a meticulously curated collection of three datasets to appraise the capabilities of generated text detection across a spectrum of domains. The Story category features narrative-rich

Table 1: SAFD demonstrates superior performance compared to competitive baselines across four adversarial test scenarios under LLM-powered style attacks, evaluated in terms of F1 Score (%). The strategies for text stylization are in Table 3. Bold denotes the overall best results.

| Method | Story | | | | PolitiFact | | | |
|---|---|---|---|---|---|---|---|---|
| | A | B | C | D | A | B | C | D |
| GLTR (ACL 2019) | 52.09 ±1.43 | 52.98 ±0.69 | 50.37 ±1.55 | 49.40 ±1.30 | 56.17 ±1.20 | 54.05 ±1.86 | 53.41 ±0.98 | 52.85 ±0.70 |
| Detectllm (Emnlp 2023) | 65.57 ±0.35 | 62.81 ±0.61 | 65.26 ±0.63 | 62.27 ±1.01 | 64.60 ±1.64 | 66.91 ±1.17 | 63.40 ±1.28 | 61.04 ±1.00 |
| LLMDet (Emnlp 2023) | 61.94 ±0.28 | 60.31 ±0.76 | 58.83 ±1.84 | 59.91 ±1.27 | 61.81 ±1.64 | 63.82 ±0.90 | 59.38 ±0.44 | 58.14 ±1.93 |
| DetectGPT (ICML 2023) | 68.92± 5.67 | 67.85± 4.42 | 62.81± 0.82 | 66.35± 2.19 | 65.83± 1.58 | 68.74± 2.71 | 72.49± 0.38 | 65.52± 1.58 |
| SeqXGPT (Emnlp 2023) | 59.63± 0.61 | 57.41 ±1.47 | 59.81 ±1.01 | 64.75 ±0.97 | 69.58 ±0.61 | 61.98 ±0.49 | 55.55 ±1.07 | 57.90 ±1.66 |
| COCO (Emnlp 2023) | 70.28 ±0.86 | 69.11 ±0.52 | 69.90 ±0.96 | 68.84 ±1.26 | 71.75 ±0.48 | 71.86 ±1.33 | 68.39 ±1.24 | 69.55 ±1.19 |
| BERT-finetuned (ACL 2024) | 51.25 ±0.49 | 54.28 ±1.67 | 55.75 ±0.93 | 52.22 ±2.24 | 55.19 ±1.01 | 56.58 ±1.28 | 57.43 ±1.17 | 55.34 ±1.92 |
| RoBERTa-finetuned (ACL 2024) | 65.28 ±1.35 | 66.69 ±2.51 | 67.47 ±1.95 | 79.00 ±1.88 | 70.32 ±1.28 | 77.00 ±2.35 | 68.37 ±1.11 | 66.52 ±1.47 |
| T5-Sentinel (Emnlp 2024) | 62.93 ±1.49 | 71.84 ±2.03 | 72.37 ±1.57 | 71.76 ±1.02 | 72.39 ±1.80 | 78.23 ±2.41 | 73.07 ±2.41 | 70.59 ±1.26 |
| OUTFOX (AAAI 2024) | 78.03 ±0.87 | 72.91 ±1.26 | 70.99 ±1.23 | 72.58 ±2.01 | 80.55 ±1.02 | 79.84 ±1.58 | 70.05 ±1.84 | 76.45 ±1.10 |
| GECScore (ACL 2025) | 70.13 ±0.48 | 77.44 ±0.58 | 76.99 ±1.16 | 67.04 ±1.09 | 63.34 ±0.95 | 83.29 ±1.53 | 67.60 ±0.32 | 67.42 ±1.56 |
| GPT-2 (2019) | 55.20± 2.42 | 56.96± 2.45 | 50.51± 1.01 | 49.40± 2.06 | 50.86± 1.93 | 52.03± 5.52 | 55.32± 3.47 | 55.89± 5.18 |
| Neo-2.7B (2020) | 61.46 ±0.95 | 55.77 ±1.84 | 58.82 ±0.40 | 69.48 ±1.27 | 71.01 ±1.61 | 62.49 ±2.08 | 56.99 ±1.80 | 59.13 ±0.82 |
| OPT-2.7B (2022) | 54.30± 0.99 | 57.29± 6.56 | 52.51± 5.36 | 51.27± 5.02 | 52.85± 3.72 | 51.53± 1.72 | 49.24± 0.58 | 48.68± 0.50 |
| LLaMA-7B (2023) | 57.56± 2.75 | 51.93± 2.63 | 51.45± 6.89 | 52.26± 0.71 | 53.12± 3.82 | 56.10± 3.12 | 56.64± 0.94 | 54.39± 4.17 |
| LLaMA2-13B (2023) | 62.60 ±1.38 | 60.61 ±0.60 | 61.69 ±1.04 | 57.14 ±0.22 | 61.64 ±1.91 | 63.60 ±0.23 | 59.07 ±1.00 | 59.92 ±1.14 |
| GPT-3.5-turbo (2023) | 72.92± 0.58 | 76.41± 0.75 | 68.68± 4.85 | 69.85± 2.63 | 69.77± 0.05 | 71.08± 0.40 | 65.50± 0.87 | 68.62± 2.53 |
| GPT-NeoX (2024) | 71.83 ±1.46 | 71.64 ±1.17 | 69.52 ±0.33 | 69.30 ±0.44 | 75.90 ±1.96 | 74.48 ±1.78 | 67.73 ±2.00 | 67.90 ±1.37 |
| GPT-4 (2024) | 73.86 ±2.06 | 70.53 ±2.32 | 75.75 ±1.46 | 78.55 ±0.39 | 77.34 ±2.33 | 79.88 ±2.20 | 68.78 ±0.18 | 71.80 ±1.65 |
| Gemma (2025) | 74.18 ±8.32 | 74.06 ±1.17 | 66.78 ±0.74 | 69.60 ±5.88 | 69.62 ±8.61 | 73.74 ±1.56 | 69.28 ±9.08 | 72.92 ±5.77 |
| Qwen-2 (2025) | 76.05 ±2.12 | 75.95 ±5.69 | 70.52 ±0.08 | 74.87 ±1.85 | 78.08 ±5.38 | 79.66 ±4.08 | 65.04 ±2.94 | 72.54 ±0.88 |
| Deepseek-R1 (2025) | 73.74 ±0.77 | 73.08 ±0.56 | 72.00 ±3.74 | 67.77 ±6.86 | 75.73 ±3.66 | 82.83 ±0.31 | 69.10 ±1.24 | 75.95 ±0.29 |
| SAFD | **79.56 ±1.28** | **78.37 ±1.99** | **77.60 ±1.24** | **79.26 ±1.08** | **81.82 ±0.68** | **84.69 ±1.35** | **79.18 ±0.95** | **79.32 ±1.18** |

texts, including literary works from the Gutenberg dataset Gerlach & Font-Clos (2020) and story-focused articles from the X-Sum dataset Narayan et al. (2018). This category assesses the model's capability to comprehend long texts and generate coherent narratives. The PolitiFact category utilizes a combined dataset from the LIAR Wang (2017) and FakeNewsNet datasets Shu et al. (2018), encompassing political statements with truthfulness ratings and additional social media context. This category assesses the model's performance in information accuracy and factual consistency. The science category encompasses scientific and domain-specific texts, such as medical records from the MedNLI dataset Romanov & Shivade (2018) or scientific literature from the Gutenberg dataset, evaluating the model's proficiency in handling specialized terminology, logical reasoning, and domain knowledge.

**Metrics.** To evaluate the detector's capability to distinguish between texts generated by LLMs and humans, especially under style-based attacks from LLMs-powered adversaries, we employ three primary performance metrics, e.g., Accuracy (A), Area Under the Receiver Operating Characteristic Curve (AU-ROC), and the F1 score (F1).

| Scenarios | Description |
|---|---|
| A | Introduce minor errors and colloquial expressions |
| B | Increase personalization and subjectivity |
| C | Mix topics and cite diverse resources |
| D | Use specialized terminology and cultural references |

Figure 3: Strategies for text stylization based on different writing styles

**Baselines.** We undertook a comparative analysis of our proposed methodology against several state-of-the-art approaches dedicated to detecting text generated by LLMs. DetectLLM Su et al. (2023) assesses text origin via log perplexity, indicating predictability. GLTR Gehrmann et al. (2019) combines statistical methods with visual analytics to highlight anomalous token probabilities, aiding in the identification of machine-generated text. DetectGPT Mitchell et al. (2023) leverages the curvature properties of language model probability functions within a probabilistic framework to identify synthetic text. LLMDet Wu et al. (2023) uses surrogate perplexity calculations tailored to each LLM, offering a model-agnostic solution for

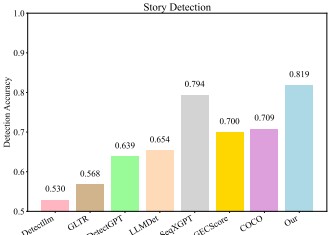 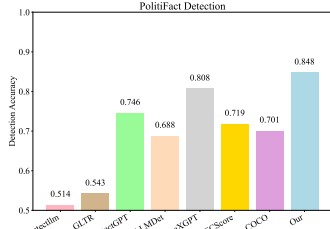 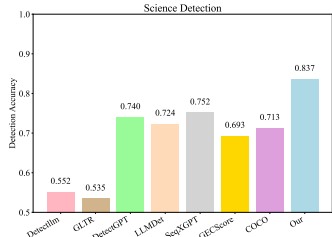

Figure 5: The visual data presented in the graphs clearly indicates that our methodology excels in detection accuracy across multiple categories,e.g., Story, PolitiFact, and Science. Our approach consistently outperforms other methods, achieving the highest accuracy in each category.

text provenance. SeqXGPT Wang et al. (2023a) represents sentences as waveforms, utilizing convolutional networks and self-attention mechanisms for sentence-level detection. GECScore Wu et al. (2024b) evaluates text similarity using a grammar error correction model, providing a robust metric for LLM-origin detection. The OUTFOX method Koike et al. (2024) bolsters the robustness of detecting text generated by LLMs by implementing an iterative in-context learning framework. COCO Liu et al. (2023) enhances detection through contrastive learning. T5-Sentinel Chen et al. (2023) employs a supervised learning approach, reframing LLM-generated text detection as a token prediction task.

**General-purpose LLMs.** LLMs perform zero-shot veracity prediction, enabling the evaluation of truthfulness without requiring task-specific fine-tuning. We use some representative baseline LLMs for analysis: GPT-2 Radford et al. (2019), OPT-2.7B Zhang et al. (2022), Neo-2.7B Gao et al. (2020), LLaMA-7B Touvron et al. (2023), LLaMA-13B, GPT-NeoX Black et al. (2022), GPT-3.5-turbo, GPT-4, Gemma Team et al. (2024), Qwen-2 Wang et al. (2024b), Deepseek-R1 Guo et al. (2025). These models serve as benchmarks to assess the capabilities and limitations of LLMs in zero-shot detecting content-generated tasks.

| Method | Story | PolitiFact | Science |
|---|---|---|---|
| OPT-2.7B | 49.74±1.19 | 52.42±2.15 | 55.77±1.97 |
| SAFD-OPT-2.7B | **64.03± 2.55** | **58.92± 2.60** | **58.58± 5.90** |
| Neo-2.7B | 59.95 ± 3.39 | 62.65 ± 3.95 | 60.43 ± 1.72 |
| SAFD-Neo-2.7B | **82.71 ± 2.04** | **80.32 ± 3.19** | **77.55 ± 1.64** |
| LLaMA2-13B | 59.99 ± 2.45 | 60.17 ± 1.43 | 58.27 ± 1.28 |
| SAFD-LLaMA2-13B | **80.23 ± 2.69** | **84.19 ± 2.38** | **86.31 ± 1.45** |
| GPT-NeoX | 61.29 ± 1.62 | 64.00 ± 2.93 | 67.38 ± 2.26 |
| SAFD-GPT-NeoX | **77.38 ± 1.66** | **83.20 ± 1.21** | **79.05 ± 2.27** |
| Qwen-2 | 66.43± 0.59 | 61.32± 0.61 | 67.49± 2.39 |
| SAFD-Qwen-2 | **85.23± 6.44** | **87.24± 3.14** | **85.57± 0.39** |

Figure 4: On different LMM backbones, SAFD demonstrates stable improvements on accuracy.

## 6.2 PERFORMANCE EVALUATION

**F1-Score.**Table 1 illustrates the performance of different methods in addressing four distinct adversarial attack styles. The results clearly show that SAFD consistently exhibits advantages across all test scenarios. In the Story scenario, SAFD achieves F1 of 79.56%, 78.37%, 77.60%, and 79.26%, respectively, with improvements of over 6.69% compared to DetectGPT, demonstrating outstanding adversarial handling capabilities. In the PolitiFact scenario, SAFD's performance is particularly remarkable, especially under colloquial adversarial attacks, where it achieves an F1 of 84.69%, surpassing COCO's 71.86% with a performance gain of nearly 12.83%. This result underscores SAFD's excellent adaptability to complex politics-related text. This further highlights SAFD's robust capability to recognize adversarial features in complex scientific texts.

**Accuracy**. As illustrated in Figure 5, our method demonstrates improvements across multiple categories, including Story, PolitiFact, and Science. Specifically, SAFD achieves detection accuracies of 0.819, 0.848,

and 0.837, respectively. When compared to other advanced methods, SAFD consistently outperforms them by margins ranging from 6.5% to 33.9%. These results highlight SAFD's superior capability in analyzing multi-dimensional textual features, effectively resisting adversarial attacks, and maintaining high precision.

Table 2: Ablation Study of SAFD Loss Components under Different attack Scenarios (F1 Score %). The strategies for text stylization are in Table 3.

| Experiment Setting | Dataset | Attack Scenario A | Attack Scenario B | Attack Scenario C | Attack Scenario D |
|---|---|---|---|---|---|
| Baseline ($L_{class}$ Only) | Story | $64.10_{\pm 1.63}$ | $63.50_{\pm 1.86}$ | $67.00_{\pm 1.53}$ | $65.90_{\pm 1.71}$ |
| | PolitiFact | $66.80_{\pm 1.95}$ | $66.20_{\pm 2.01}$ | $69.56_{\pm 1.76}$ | $68.30_{\pm 1.83}$ |
| | Science | $62.04_{\pm 1.47}$ | $61.57_{\pm 1.59}$ | $64.87_{\pm 1.34}$ | $63.54_{\pm 1.24}$ |
| $L_{class} + L_{style}$ | Story | $71.51_{\pm 1.33}$ | $73.22_{\pm 1.49}$ | $73.08_{\pm 1.27}$ | $72.39_{\pm 1.39}$ |
| | PolitiFact | $74.15_{\pm 1.63}$ | $76.59_{\pm 1.61}$ | $76.37_{\pm 1.42}$ | $75.47_{\pm 1.68}$ |
| | Science | $69.48_{\pm 1.16}$ | $71.84_{\pm 1.12}$ | $70.79_{\pm 1.53}$ | $69.80_{\pm 1.10}$ |
| $L_{class} + L_{attr}$ | Story | $73.29_{\pm 1.58}$ | $72.37_{\pm 1.14}$ | $75.06_{\pm 1.70}$ | $74.34_{\pm 1.91}$ |
| | PolitiFact | $76.68_{\pm 1.25}$ | $75.51_{\pm 1.67}$ | $79.28_{\pm 1.49}$ | $78.33_{\pm 1.96}$ |
| | Science | $70.86_{\pm 1.13}$ | $69.89_{\pm 1.40}$ | $73.75_{\pm 1.62}$ | $72.20_{\pm 1.67}$ |
| Full Model (SAFD) ($L_{class} + L_{style} + L_{attr}$) | Story | $78.45_{\pm 1.12}$ | $78.00_{\pm 1.91}$ | $79.50_{\pm 1.50}$ | $79.08_{\pm 1.34}$ |
| | PolitiFact | $80.69_{\pm 1.79}$ | $82.58_{\pm 1.35}$ | $81.88_{\pm 1.02}$ | $81.43_{\pm 1.08}$ |
| | Science | $84.92_{\pm 1.15}$ | $80.47_{\pm 1.62}$ | $78.57_{\pm 1.49}$ | $83.31_{\pm 1.54}$ |

**Different Backbones**. The SAFD method demonstrates performance improvements across multiple models and datasets. For instance, SAFD-Neo-2.7B achieves the accuracy of 82.71% on the Story dataset, compared to 59.95% for the baseline, while SAFD-LLaMA2-13B reaches 84.19% on PolitiFact, up from 60.17%. Using the Qwen-2 backbone, the accuracy improved from 66.43% to 85.23% for Story datasets. These results highlight SAFD's ability to enhance detection accuracy by leveraging style-agnostic feature extraction and adversarial data augmentation, which ensures robustness against style-based adversarial attacks. Its style-agnostic feature extraction forces the model to learn the intrinsic, content-centric artifacts of LLM generation, thereby mitigating the risk of overfitting to superficial and easily manipulated stylistic cues. The adversarial data augmentation proactively hardens the detector by exposing it to a diverse array of synthesized edge cases that mimic sophisticated evasion attempts.

**Ablation study.** As shown in Figure 2, comparing ($L_{class} + L_{style}$) with Baseline, there's a clear increase in F1 scores across all datasets and all attack scenarios. On the Story dataset under scenario A, F1 improves from 64.10% to 71.51%. This strongly validates the effectiveness of the style alignment loss ($L_{style}$). By enforcing consistent predictions for stylistically varied content, it demonstrably enhances the detector's robustness against various rewriting techniques (colloquialism, subjectivity, topic mixing, specialized terminology). The result demonstrates a synergistic effect between $L_{style}$ and $L_{attr}$. While each auxiliary loss improves performance individually, combining them in the full SAFD framework yields the best results.

## 7 CONCLUSION

We address critical vulnerabilities in detecting LLM-generated text, particularly against style-based adversarial attacks powered by LLMs. Existing detectors exhibit performance degradation when confronted with stylistic manipulations, highlighting the limitations of their reliance on stylistic cues. To overcome these challenges, we proposed a robust detection framework that prioritizes content-driven features and employs a style-agnostic training paradigm. By leveraging adversarially enriched datasets and advanced representation learning techniques, our approach disentangles semantic content from stylistic variations, ensuring enhanced robustness against diverse adversarial attacks. Future work could explore integrating multi-modal signals or developing certified defense mechanisms to provide formal guarantees.

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

## A  APPENDIX

### A.1  THE USE OF LARGE LANGUAGE MODELS

In preparing this work, Large Language Models (LLMs) were used solely for translation and language polishing. All content, arguments, and conclusions are entirely my own, and the use of LLMs did not contribute to the generation of original ideas or substantive material.

### A.2  ADDITIONAL EXPERIMENTAL RESULTS

**AUROC**. As shown in Figure 6, our method demonstrates significant advantages in distinguishing between human-written content and content generated by LLMs. Specifically, in the Story category, SAFD achieved an AUROC value of 0.85, ranking first among all comparative methods. This result highlights its exceptional performance in handling creative and narrative texts. Furthermore, in the Science category, SAFD also performed impressively with an AUROC value of 0.84, which is not only significantly higher than that of COCO (0.78) but also surpasses GECScore (0.73). SAFD exhibits unique strengths when confronted with adversarial sample attacks across various text styles. Whether in literary creation, scientific discourse, or other types of texts, SAFD effectively resists the impact of adversarial samples, maintaining high-precision discrimination capabilities. By thoroughly analyzing multiple dimensions of textual features, including intrinsic content, word preferences, and logical coherence, SAFD can accurately identify and adapt to stylistic variations in texts, thereby providing reliable and precise judgments.

**F1 Score**. The results on the Science dataset clearly demonstrate the superior and robust performance of our proposed method, SAFD, across four distinct evaluation scenarios (A, B, C, and D). Our method achieves the highest F1 scores in three of the four scenarios, posting scores of **80.52% in A**, **76.61% in B**, and **77.55% in D**. This consistently high performance underscores its effectiveness in detecting machine-generated text under various conditions. When compared to established baselines, SAFD shows a significant advantage. For instance, in scenario A, it outperforms the next best method, OUTFOX, by over 2.4 percentage points. Similarly, in scenario D, it surpasses the second-best performer, Qwen-2, by a margin of nearly 2.7 points.

Table 3: F1 Score (%) performance on the Science dataset.

| Method | Science | | | |
|---|---|---|---|---|
| | A | B | C | D |
| GLTR (ACL 2019) | 53.86 ± 0.98 | 53.41 ± 0.94 | 50.51 ± 0.84 | 53.02 ± 1.16 |
| Detectllm (Emnlp 2023) | 63.13 ± 0.12 | 63.95 ± 1.41 | 63.89 ± 0.47 | 61.65 ± 0.56 |
| LLMDet (Emnlp 2023) | 59.35 ± 0.51 | 59.77 ± 0.69 | 59.18 ± 1.86 | 59.25 ± 0.47 |
| DetectGPT (ICML 2023) | 63.24 ± 0.17 | 64.83 ± 2.43 | 67.99 ± 6.27 | 64.01 ± 3.05 |
| SeqXGPT (Emnlp 2023) | 56.70 ± 1.59 | 59.56 ± 1.54 | 57.20 ± 0.67 | 59.35 ± 1.80 |
| COCO (Emnlp 2023) | 68.76 ± 0.31 | 68.70 ± 1.44 | 65.01 ± 1.33 | 67.43 ± 1.12 |
| BERT-finetuned | 53.27 ± 1.74 | 50.09 ± 1.81 | 52.61 ± 1.13 | 54.77 ± 1.66 |
| RoBERTa-finetuned | 76.33 ± 0.75 | 69.43 ± 1.59 | 70.30 ± 1.42 | 69.43 ± 1.56 |
| T5-Sentinel | 72.98 ± 1.05 | 71.07 ± 1.59 | 78.91 ± 2.13 | 74.42 ± 1.32 |
| OUTFOX (AAAI 2024) | 78.08 ± 1.33 | 74.57 ± 1.24 | 73.59 ± 1.58 | 70.87 ± 1.41 |
| GECScore (ACL 2025) | 66.59 ± 1.92 | 66.99 ± 0.73 | 66.78 ± 1.97 | 68.28 ± 1.99 |
| GPT-2 (2019) | 50.88 ± 0.75 | 52.06 ± 3.53 | 54.72 ± 4.46 | 50.23 ± 2.20 |
| Neo-2.7B (2020) | 59.28 ± 0.72 | 57.75 ± 1.04 | 58.19 ± 0.32 | 59.32 ± 0.90 |
| OPT-2.7B (2022) | 45.73 ± 2.83 | 54.82 ± 2.61 | 50.83 ± 2.33 | 54.64 ± 1.72 |
| LLaMA-7B (2023) | 51.52 ± 3.60 | 51.21 ± 2.89 | 59.45 ± 2.57 | 52.10 ± 3.25 |
| LLaMA2-13B (2023) | 57.85 ± 0.29 | 59.41 ± 0.31 | 60.22 ± 0.29 | 58.50 ± 0.98 |
| GPT-3.5-turbo | 68.98 ± 1.79 | 74.82 ± 2.75 | 71.90 ± 0.08 | 71.15 ± 3.92 |
| GPT-NeoX (2024) | 71.51 ± 1.33 | 74.16 ± 1.30 | 71.30 ± 1.95 | 68.62 ± 1.84 |
| GPT-4 (2024) | 75.22 ± 3.82 | 74.12 ± 3.26 | 79.15 ± 2.81 | 71.24 ± 4.44 |
| Gemma (2025) | 66.80 ± 3.91 | 75.65 ± 4.88 | 73.77 ± 4.31 | 74.34 ± 0.07 |
| Qwen-2 (2025) | 73.49 ± 3.53 | 74.72 ± 0.59 | 74.83 ± 1.97 | 74.88 ± 1.24 |
| Deepseek-R1 (2025) | 73.57 ± 0.54 | 74.11 ± 6.20 | 73.57 ± 0.44 | 70.28 ± 4.72 |
| SAFD | **80.52 ± 1.84** | **76.61 ± 1.20** | **76.80 ± 1.81** | **77.55 ± 1.64** |

Table 4: Across different sets of reframing prompts, SAFD demonstrates stable and significant improvements over the most competitive baseline on accuracy.

| Method | Story | PolitiFact | Science |
|---|---|---|---|
| Baseline (Best) | 79.14 ± 1.83 | 83.20 ± 1.48 | 75.37 ± 3.18 |
| SAFD | **82.24 ± 3.09** | **80.99 ± 2.17** | **83.50 ± 1.06** |
| $w/P_1$ | 79.95 ± 1.51 | 82.08 ± 2.13 | 83.30 ± 2.32 |
| $w/P_2$ | 78.95 ± 2.16 | 83.55 ± 1.54 | 81.66 ± 3.51 |
| $w/P_3$ | 81.95 ± 2.79 | 79.98 ± 1.29 | 81.83 ± 1.66 |
| $w/P_4$ | 80.21 ± 2.28 | 83.71 ± 1.31 | 81.97 ± 3.14 |

**Different Prompts.** The SAFD method demonstrates superior precision across multiple datasets (Story, PolitiFact, Science) compared to the best baseline, as evidenced by the experimental results in Table 4. SAFD not only delivers a notable precision of 82.24% on the Story dataset, eclipsing the baseline's 79.14%, but also demonstrates a commanding lead on the more challenging Science dataset, achieving 83.50% precision against a mere 75.37% for the baseline. The underlying drivers of this performance leap are twofold. First, its style-agnostic feature extraction paradigm allows the model to transcend superficial stylistic fingerprints, which often confound conventional detectors, and instead learn the fundamental, intrinsic signatures of synthetic text. Second, our adversarial data augmentation strategy proactively immunizes the model

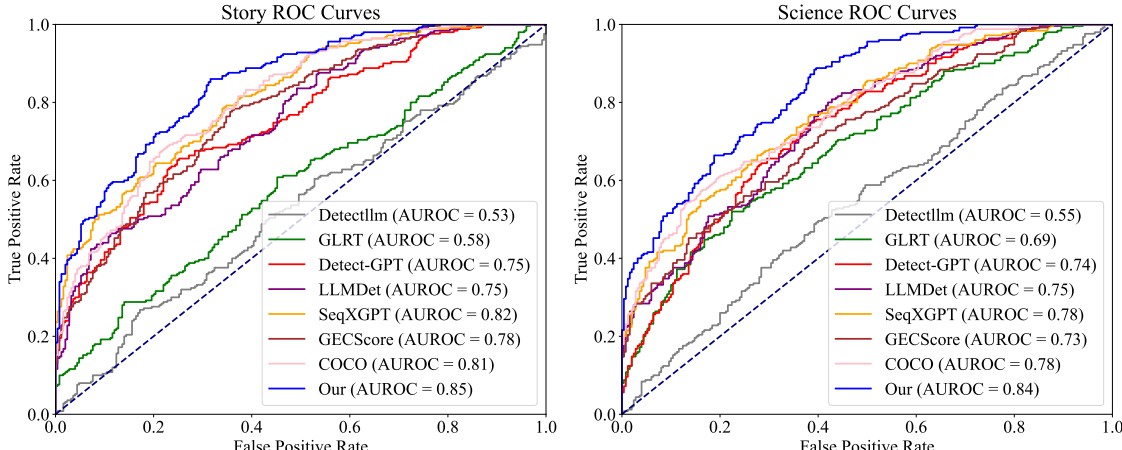

Figure 6: The figure presents the ROC curves for various methods evaluated on two categories: Story and Science. The AUROC is used to quantify each method's ability to distinguish between human and LLM-generated content.

against evasive maneuvers by training it on a curated corpus of hard-to-classify, style-manipulated examples. This synergy culminates in exceptional robustness against a broad spectrum of style-based adversarial attacks. Crucially, SAFD's high precision is consistently maintained across varied generative prompts (e.g., $w/P_1, w/P_2, w/P_3$), affirming its operational reliability and positioning it as a highly effective solution for identifying LLM-generated content. For instance, on the Story dataset, SAFD achieves a precision of 82.24, significantly outperforming the baseline's 79.14, while on the Science dataset, SAFD attains a precision of 83.50, compared to the baseline's 75.37. This improvement is attributed to SAFD's style-agnostic feature extraction and adversarial data augmentation, which enhances its robustness against style-based adversarial attacks. SAFD maintains high precision across various prompts and configurations (e.g., $w/P_1$, $w/P_2$, $w/P_3$), showcasing its adaptability and reliability in detecting LLM-generated content.

SCOPE OF CLAIMS:

The methods were primarily tested on a few specific datasets (such as Story, PolitiFact, and Science) and only run a few times (specifically, our program was run 3 times). This implies that the generalizability of the results may be somewhat limited. To comprehensively evaluate the method's effectiveness, further testing should be conducted on more diverse datasets and language environments, with an increased number of experimental runs to obtain more stable error estimates.

A.3   USE SCIENTIFIC ARTIFACTS

During our research, we used several scientific artifacts, including datasets, methodologies, and evaluation metrics, which are essential for assessing the robustness of detection systems against style-based adversarial attacks.

### A.3.1 DATASETS:

We employed a meticulously curated collection of three datasets to appraise the capabilities of generated text detection rigorously:

- **Story Category:** This category features narrative-rich texts, including literary works from the Gutenberg dataset Gerlach & Font-Clos (2020) and story-focused articles from the X-Sum dataset Narayan et al. (2018). It evaluates the model's capability to comprehend long texts and generate coherent narratives.

- **PolitiFact Category:** Utilizes a combined dataset from the LIAR Wang (2017) and FakeNewsNet datasets Shu et al. (2018), encompassing political statements with truthfulness ratings and additional social media context. This assesses the model's performance in information accuracy and factual consistency.

- **Science Category:** Encompasses scientific and domain-specific texts, such as medical records from the MedNLI dataset Romanov & Shivade (2018) or scientific literature from the Gutenberg dataset, evaluating the model's proficiency in handling specialized terminology, logical reasoning, and domain knowledge.

### A.3.2 METHODOLOGIES:

Our research leverages intermediate layer outputs and weight parameters of LLMs to differentiate between human-authored and LLMs-generated texts Gehrmann et al. (2019); Mitchell et al. (2023); Wu et al. (2023); Wang et al. (2023a); Wu et al. (2024b); Liu et al. (2023).

We introduced subtle text modifications to monitor changes in log probabilities, where a notable decrease typically indicates LLM-generated text, while an increase or minor fluctuations suggest human authorship. To detect text generated by LLMs, statistical analyses are employed to examine features such as word choice, sentence structure, and stylistic elements Taguchi et al. (2024); Wang et al. (2023b); Bao et al. (2023); Su et al. (2023); Bakhtin et al. (2019).

### A.3.3 EVALUATION METRICS:

To evaluate the detector's capability to distinguish between texts generated by LLMs and humans, especially under style-based attacks, we employ three primary performance metrics:

- **Accuracy (A):** Measures the proportion of correctly classified instances.

- **Area Under the Receiver Operating Characteristic Curve (AUROC):** Evaluate the trade-off between true and false positive rates.

- **F1 Score (F1):** Harmonic mean of precision and recall, providing a balanced performance measure.

### A.3.4 ADVERSARIAL TEST SCENARIOS:

We systematically assessed the robustness of detection systems against style-oriented adversarial attacks using transformed test samples. For example:

- Narrative texts were rewritten using the writing style of Andersen's fairy tales.

- Political texts were adapted to the style of CNN.

- Scientific texts were modified to match the writing style of Nature and Science.

These scientific artifacts form the backbone of our research, enabling us to conduct rigorous evaluations and draw meaningful conclusions about the effectiveness of various detection methods against style-based adversarial attacks.

## A.4 REPRODUCIBILITY STATEMENT

### A.4.1 CODE AND DATA AVAILABILITY

The dataset and code used in this study are available at an anonymized link [2]. SAFD is designed to prioritize content-driven features over stylistic cues to improve robustness against adversarial attacks. The framework's training objective combines three loss functions: a style alignment loss ($L_{\text{style}}$), a classification loss ($L_{\text{class}}$), and a content-focused veracity attribution loss ($L_{\text{attr}}$). The methodology involves augmenting training data by transforming human-written texts into multiple stylistic variants using LLM.

### A.4.2 HARDWARE DEVICES

All our experiments were meticulously conducted on a high-performance computing platform running Ubuntu. The platform is powered by an Intel(R) Xeon(R) Platinum 8176 CPU @ 2.10GHz, delivering robust computational capabilities. The system is equipped with a substantial 503 GB of memory, ensuring efficient data processing and storage. Additionally, to further enhance computational power, we utilized four NVIDIA Corporation GA102GL RTX A6000 GPUs. These GPUs provided the necessary parallel processing power to handle the intensive computational tasks associated with our research. The stability and broad support of the Ubuntu operating system allowed us to fully leverage the hardware's performance, ensuring the smooth execution of experiments and the reliability of our results.

## A.5 DATASETS

Our experimental framework utilizes a carefully curated set of datasets to evaluate the capabilities and limitations of Large Language Models (LLMs) across various domains and scenarios. These datasets are meticulously selected to reflect the diverse range of content types that LLMs might encounter.

Specifically, we use three primary datasets for rigorous evaluation of text generation and detection capabilities:

- **Story Dataset**: Contains narrative texts, used to assess model performance on story-like content.
- **PolitiFact Dataset**: Includes politically-oriented texts, utilized to test the model's effectiveness on complex political content.
- **Science Dataset**: Consists of scientific texts, employed to evaluate the model's performance on technical and specialized content.

For each LLM, we randomly draw samples from these datasets and perform detailed analyses on the generated texts. The generated output is divided into two parts:

- 15,000 samples for statistical analysis, tracking the linguistic features.
- 15,000 samples for validation purposes, ensuring robustness and accuracy of the models.

These validation sets are combined with 15,000 human-written texts sourced from specific datasets, forming a unified corpus used for training and validating text detectors. This approach allows us to systematically compare LLM-generated texts with human-written texts.

---

[2]https://anonymous.4open.science/status/A-Style-Agnostic-Framework-for-Detecting-LLM-Generated-Text-90B7

To further test the model's robustness against shifts in data distribution, we employ adversarial attacks based on different writing styles, such as those characteristic of Andersen's fairy tales and prestigious academic journals like Nature and Science. For instance, modern narratives might be rewritten with whimsical language, moral undertones, and vivid imagery typical of fairy tales. Our general prompt format for these transformations is structured as follows:

> **Rewrite the following text using the style of [publisher/book]:** [input text]

For narrative texts, we adopt the writing style of Andersen's fairy tales. For political texts, we use the style of CNN, and for scientific texts, we utilize the style of Nature and Science. These stylistically altered test samples, transformed using LLM-based techniques, are utilized for the systematic assessment of detection systems' robustness against style-oriented adversarial attacks.

### A.5.1 METRICS

To ensure the accuracy and reliability of the results, each experiment was conducted in triplicate, and the standard deviations were calculated. This approach effectively assesses the stability and consistency of the data, thereby enhancing the credibility of our conclusions. To assess the detector's capability to differentiate between texts generated by large language models (LLMs) and those written by humans, we utilize Accuracy (A) and the Area Under the Receiver Operating Characteristic Curve (AUROC) as primary performance metrics. Additionally, we consider other metrics, such as F1 scores (F1) and Recall (R), to provide a more comprehensive evaluation.

$$A = \frac{TP + TN}{TP + TN + FP + FN} \tag{11}$$

$$R = \frac{TP}{TP + FN}; \quad F1 = 2 \times \frac{P \times R}{P + R} \tag{12}$$

True Positives ($TP$) refer to human-written texts correctly identified by the model. True Negatives ($TN$) represent texts generated by LLMs accurately classified as LLMs-generated. False Positives ($FP$) denote LLMs-generated texts incorrectly labeled human-written, while False Negatives ($FN$) correspond to human-written texts the model fails to identify correctly.

