# OpenReview forum: "SAFD: A Style-Agnostic Framework for Detecting LLM-Generated Text"
_ICLR.cc/2026/Conference — ICLR 2026 Conference Withdrawn Submission_

### Official Review · Reviewer_umyP · 2025-10-16

**Soundness:** 2
**Presentation:** 1
**Contribution:** 2
**Rating:** 2
**Confidence:** 4

**Summary:**

The paper targets the vulnerability of machine-generated text (MGT) detectors to style-based attacks and proposes SAFD, a style-agnostic training framework that encourages detectors to rely on content rather than superficial stylistic cues. SAFD augments both human and machine texts by rewriting them into multiple styles with a large language model (e.g., newswire, academic, colloquial), then optimizes a tri-objective loss: a standard classification loss, a style-consistency loss that aligns predictions across the style variants via pairwise KL divergence, and an attribution-guided loss that uses auxiliary LLM signals to supervise content-level indicators. Across Story, PolitiFact, and Science domains and several style-attack regimes, SAFD achieves consistent improvements over strong baselines such as DetectGPT, GLTR, COCO, and others, and the gains hold across diverse detector backbones (OPT, GPT-Neo, LLaMA, Qwen, NeoX). Ablations substantiate the contribution of each component, and the authors commit to releasing code/data; remaining gaps include broader attack families and multilingual evaluation.

**Strengths:**

1. Extensive empirical results and comparative experiments.
2. Appears to address a timely and important problem.
3. Clear and well-defined problem statement.

**Weaknesses:**

This paper is not well written; it is rife with inconsistent use of key terminology and citation errors (see weaknesses below), along with many empirically stated claims that are insufficiently substantiated. The core motivation is also unconvincing. Below I list the deficiencies in order of severity:

1. In the paper you claim as a core motivation that “State-of-the-art detectors for generated text are found to be impacted by LLM-driven stylistic variations.” However, the table used to support this claim—Table 1 (which the authors mistakenly label as Figure 1)—lists detectors such as GLTR (2019), DetectGPT (2023), and LLMDet (2023), which cannot reasonably be considered current SOTA. Moreover, your prompts for generating adversarial examples take the form: “Rewrite the following text using the style of [publisher/book]: [input text],” which is essentially a paraphrasing attack. Many recent studies indicate that paraphrasing attacks alone can substantially reduce detection accuracy. It is therefore unclear whether the observed drop is due to paraphrasing per se or to stylistic transfer. In other words, if one simply used the prompt “Rewrite the following text: [input text],” how would the accuracy change?

2. Several claims are insufficiently supported. For example, at line 38 you write “Many state-of-the-art detection methods heavily rely on stylistic attributes...,” yet recent SOTA baselines such as Fast-DetectGPT and Binoculars do not appear to rely on stylistic attributes. In addition, at lines 61 and 65 you repeatedly state that “this is the first...,” which seems overstated and likely exaggerates the paper’s contribution.

3. Critical implementation details are vague. In the paper (and in the appendix, which I also checked) I could not find details of your detector implementation (e.g., the base model behind the Table 1 results, learning rate, batch size, etc.). Furthermore, what are P1-P4 in Table 4? The paper appears never to define them.

4. There are numerous spelling and citation errors. An immediate impression is that the paper confuses `\citep` and `\citet` in many places (e.g., lines 35, 38, 39, etc.), and you claim that GECScore is published in ACL 2025 in Table 1, line 294, but this paper was published in COLING 2025. There are also multiple table/figure mislabelings—for example, Figure 1 should clearly be Table 1, and Figure 4 should be Table 4. In addition, the text in Figure 5 is too small to read. Finally, what is “LMM” in Figure 2 and Figure 4? Is it a typo for “LLM,” or does it mean something else (e.g. **L**arge **M**ultimodal **M**odel, but I don't think so, because this is a text classification task)?

5. Though this paper **does** contains many baselines, it still misses some important baselines like ImBD [a].

[a] Chen, Jiaqi, et al. "Imitate Before Detect: Aligning Machine Stylistic Preference for Machine-Revised Text Detection." Proceedings of the AAAI Conference on Artificial Intelligence. Vol. 39. No. 22. 2025.

Given the presentation and quality of this paper, I recommend a clear rejection of this paper in its current form. I hope the author can take the suggestions above into consideration in the future form.

**Questions:**

See weaknesses

---

### Official Review · Reviewer_stn8 · 2025-10-31

**Soundness:** 3
**Presentation:** 4
**Contribution:** 3
**Rating:** 6
**Confidence:** 5

**Summary:**

This paper addresses the vulnerability of LLM-generated text detectors to style-based adversarial attacks. They propose SAFD (Style-Agnostic Framework for Detection), which aims to detect machine-generated text by focusing on content-driven features rather than stylistic attributes. They demonstrate that state-of-the-art detectors suffer significant accuracy drops under style-based attacks. Based on this, they introduce a novel training framework combining style alignment loss, classification loss, and content-focused attribution loss, and construct adversarially enriched datasets using LLMs fine-tuned for diverse style attacks.

**Strengths:**

1. The systematic investigation of LLM-driven stylistic variations on detection system performance is timely and relevant, addressing an important gap in the adversarial robustness literature.
2. The three-component loss function represents a reasonable integration of complementary objectives for style-invariant learning.
3. The experimental evaluation is reasonably comprehensive, testing across multiple datasets (Story, PolitiFact, Science) and comparing against numerous baselines, including both specialized detectors and different LLMs.

**Weaknesses:**

1. The authors acknowledge running each experiment only three times, which may be insufficient for robust statistical conclusions. Moreover, in Tables 1 and 3, the standard deviations are occasionally high (up to 9.08), raising concerns about result stability and reproducibility.

2. Including an analysis of computational cost or training time compared to the baselines would be very helpful. Such information would allow readers to better assess the practical efficiency and scalability of the proposed method.

3. Section 5.4 relies on querying an LLM with a predefined feature set, but the reliability of these queried features is not justified. A brief validation or sensitivity analysis of this step would strengthen confidence in the reported findings.

4. Symbols such as 𝑆 and 𝐴  appear in Figure 2 but are never defined in the text, making it difficult to interpret the figure accurately. These should be clearly explained, either in the caption or in the main body.

**Questions:**

1. How are the 15,000 samples selected in lines 791-795? What is the train/test split?
2. Key implementation details are missing: learning rates, batch sizes, and number of epochs.

---

### Official Review · Reviewer_Ue1r · 2025-10-31

**Soundness:** 2
**Presentation:** 3
**Contribution:** 2
**Rating:** 2
**Confidence:** 4

**Summary:**

This paper explores the effect of stylistic re-writes on machine-text detection performance, finding that such re-writes lead to a general decline in the performance of standard detectors. Motivated by this, the paper proposes SAFD, a “style-agnostic” training recipe that enhances the robustness of detectors in face of stylistic re-writes. The SAFD approach introduces two new loss functions in addition to the standard binary cross-entropy for classification. The first loss function ensures that stylistic variants of the same content have similar predictions via KL divergence, and the second predicts various predictive features of text generated by LLMs.

**Strengths:**

* S1 - Presents an interesting training routine, in particular the style alignment loss and specially the content focused attribution loss are interesting additions to the training pipeline.
* S2- Although others have looked at re-writes before, it’s interesting to note that this paper is the first to my knowledge to look at stylistic re-writes.

**Weaknesses:**

* W1 - In Figure 1, the semantic similarity of the re-writes to the original is not shown. Are the re-writes destroying content as well as style? And if so, could it be that the adversarial attack isn’t just modifying the style?
* W2 - It has become standard for detection works to evaluate the AUROC at low FPR (1%) settings, where detectors must do well in order to be used in real-world scenarios. See works such as: https://arxiv.org/pdf/2405.07940, https://arxiv.org/pdf/2401.12070 and https://arxiv.org/pdf/2401.06712
* W3 - Some important, popular, and well performing detectors are missing, in particular Binoculars and FastDetectGPT: https://arxiv.org/abs/2401.12070, https://arxiv.org/abs/2310.05130
* W4 - There are many standard machine-text detection benchmarks that could’ve been used such as RAID and MAGE: https://arxiv.org/abs/2405.07940, https://arxiv.org/abs/2305.13242. MAGE in particular would’ve been helpful as it provides many different test-beds that control for things like unseen LLMs during testing time.
* W5 - Insufficient robustness analysis. For example, what happens when SAFD is trained on one set of LLMs and evaluated on another? How about when it’s trained on one domain and evaluated on another? A lot of these testbeds are already provided in the MAGE dataset as stated above.

**Questions:**

In general, I'm mainly concerned about W5 / W4 and W3. Addressing these weaknesses would improve upon my score.

---

### Official Review · Reviewer_R89W · 2025-11-01

**Soundness:** 2
**Presentation:** 1
**Contribution:** 2
**Rating:** 2
**Confidence:** 3

**Summary:**

This paper introduces SAFD, a style-agnostic detection framework that focuses on content-based features to improve robustness against style-based adversarial attacks. Motivated by their observations that existing detectors degrade their accuracy when LLMs manipulate writing style, the authors propose a style-invariant training paradigm and use adversarially enriched datasets to disentangle content from style. Experiments across multiple datasets demonstrate that SAFD significantly enhances both accuracy and robustness compared to existing detectors.

**Strengths:**

- A new detection approach motivated by the observation that recent LLMs can generate texts in a wide variety of writing styles, suggesting that content-based detection rather than style-based detection is more appropriate.
- Extensive evaluation experiments across a wide range of baseline detectors, as well as ablation studies to analyze the effectiveness of different components of their approach.

**Weaknesses:**

- Overall, the paper feels somewhat poorly written, and the following unclear points make the results appear unreliable:
    - It is not specified how the threshold for determining accuracy (or classification) was chosen.
    - It is unclear which models were used to generate the LLM side of the detection dataset, as well as the corresponding generation settings. For example, were the texts generated from what prompts and hyper-parameters?
    - Figures 3 and 4 are never referenced in the main text. How exactly was the style variation in Figure 3 prompted to the LLM?
    - When using general-purpose LLMs as detectors, how was the detection actually performed? What prompts were used?
- Regarding the claim that the paper is *“the first to systematically investigate the impact of these LLM-driven stylistic alterations on the performance of text detection systems”*, it is questionable because some similar work already exist [1,2]
- As for Figure 5, my understanding is that it corresponds to the original (non-adversarial) setting rather than the style-altered one. If so, why does it not include OUTFOX, which often showed runner-up performance in Table 1?

---

References:

[1] Shi et al. Red Teaming Language Model Detectors with Language Models. TACL 2023.

[2] Koike et al. How You Prompt Matters! Even Task-Oriented Constraints in Instructions Affect LLM-Generated Text Detection. EMNLP Findings 2024.

**Questions:**

See the weaknesses part

---

### Note · Authors · 2025-11-14

I have read and agree with the venue's withdrawal policy on behalf of myself and my co-authors.